# A Complex Characterization of Pumpkin and Quince Purees Obtained by a Combination of Freezing and Conventional Cooking

**DOI:** 10.3390/foods11142038

**Published:** 2022-07-09

**Authors:** Oana Viorela Nistor, Gabriel Danut Mocanu, Doina Georgeta Andronoiu, Viorica Vasilica Barbu, Liliana Ceclu

**Affiliations:** 1Faculty of Food Science and Engineering, Dunarea de Jos University of Galati, 800201 Galati, Romania; oana.nistor@ugal.ro (O.V.N.); danut.mocanu@ugal.ro (G.D.M.); georgeta.andronoiu@ugal.ro (D.G.A.); vasilica.barbu@ugal.ro (V.V.B.); 2Faculty of Economics, Engineering and Applied Siences, Cahul State University “Bogdan Petriceicu Hasdeu”, 3901 Cahul, Moldova

**Keywords:** pumpkin, quince, puree, bioactive compounds, freeze, conventional cooking

## Abstract

Two main sources of fibers and bioactive compounds represented by pumpkin (*Cucurbita maxima* L.) and quince *(Cydonia oblonga Mill.)* were selected for the present study. The current changes in consumers behavior oblige manufacturers to develop new assortments of ready-to-eat products, considering their nutritional characteristics. Hence, this study aimed to process free sugar pumpkin and quince puree using a combination of freezing (−15 °C) and cooking at 95 °C for 20 min. Four variants of purees were obtained by using different combinations between pumpkin and quince (pumpkin puree, quince puree, and pumpkin and quince puree in ratios of 1:1 and 3:1). The samples were characterized in terms of complex interconnected analysis, which could provide further information for the added-value products. Thus, highest values of β-carotene content were attributed to pumpkin puree (P −5.34 ± 0.05 mg/g DW) and pumpkin and quince puree 3:1 (PQ 3:1 −3.78 ± 0.014 mg/g DW). These findings are also supported by the values of ABTS inhibition, which was registered as 71.32% for the P sample and 76.25% for the PQ 3:1 sample. The textural analysis revealed firmness values of 1.27 N for pumpkin puree and 2.33 N for quince puree. Moreover, the structural changes were minimum, while the cellular structure and some tissues were preserved intact.

## 1. Introduction

At present, one of the most important concerns of the food security and assurance domain is represented by illnesses in people, which are considerably increasing. Obesity, diabetes, cancer, cardiovascular, and other chronic diseases are mostly correlated to the reduced consumption of natural sources of vitamins and the increased intake of several other unhealthy compounds. 

In addition to the problem of an unbalanced diet, these diseases are the result of the fast pace of life of the contemporary world [1]. Thus, the need to improve human health should be a continuous pursuit of the researching and industrial field.

The consumption of nutritive ready-to-eat products as pureed fruits and vegetables has become widespread throughout the world, due to the reduction in storage and transport costs and the possibility of creating a reserve of “fruits and vegetables” for the off-season. 

Fruit and vegetable purees are recommended in various special diets (senior/baby food, people suffering from dysphagia, weight loss treatments, low-sugar, hyposodic, or dyslipidemia diets, and so on). Fruits purees are appreciated by the consumers due to their pleasant sensory properties (color, flavor, taste, and texture) and their biologically active compounds, such as total phenols, vitamins, micro and macronutrients, and dietary fibers [2]. 

Overall acceptability and other sensorial properties are important in consumer decisions; thus, it is important to choose proper fruits to satisfy both the hedonistic and the nutritional aspects. 

Among the most requested fruits is pumpkin due to its high agricultural productivity worldwide, cheapness, high added value, and various possibilities of cooking [3]. Pumpkin is of great interest in human nutrition due to the high amounts of carotenoids (β-carotene, lutein, and lycopene), polyphenols, flavonoids, pigments, polysaccharides, sterols, proteins, peptides, macro elements, micro elements (iron, iodine, zinc, copper, and selenium), and vitamins [4,5].

The processing methods can vary, e.g., soups, pies, puree, juice, marmalade, and jam. Alone or in various combinations, pumpkin is considered a very valuable raw material.

Currently, there are a wide range of pumpkin purees combined with other fruits or vegetables because of their healthy and nutritious properties. In addition, it is a highly preferred fruit because of its meaty and non-sandy texture, providing smoothness to the food [6]. 

An exceptional combination consists of pumpkin and quince, due to the sweetness and smoothness of pumpkin together with the astringency and flavor of quince.

Fresh quinces known for their astringency, sourness, and hardness are not always a consumer choice.

Therefore, they are usually processed into other forms such as sweetened juice, compote, puree, marmalade, jam, jelly, cake, or liquor, which are more appreciated in many countries, due to the pleasant, lasting, and powerful flavor and high content of dietary fibers (especially pectin) [7]. Ripe quince fruits have a strong floral odor, which is due to the presence of the essential oils, which are mostly located in the peel. Some compounds responsible for this flavor are formed during technological processing [8].

The association between pumpkin and quince is considered exceptional due to the fact that it does not require the addition of preservatives and sugar. 

Unsweetened pumpkin and quince puree can be consumed as such or used as intermediate products, allowing the consumers worldwide to adjust the meal to their personal needs. It can be used in pastries, ice creams, sorbets, jam, and marmalade or in the beverage industry to create original cocktails, beers, and other drinks [3].

The present study proposes a novel combination of technologies based on freezing and thermal processing to obtain free sugar pumpkin and quince puree. The importance and novelty of the study is sustained by the accordance with the international regulations regarding the reduction in sugar intake among adults and children [9], by using fruit products as ingredients in foods instead of refined sugars [10].

Freezing and thermal treatments are among the most important methods to preserve foods. Freezing softens plant tissue and thermal treatment inactivates deteriorative microorganisms and enzymes, in addition to improving the bioavailability [11]. 

In this study, four variants of pumpkin and quince puree (in different combinations) were investigated, and the impact of selected technologies on the phytochemical, textural, rheological, color, sensorial, and microstructural properties of purees were evaluated.

## 2. Materials and Methods

### 2.1. Chemicals 

Ethanol, Folin–Ciocâlteu’s reagent, sodium carbonate, ABTS (2,2-azino-bis(3-ethylbenzothiazoline-6-sulfonic acid) diammonium salt), quercetin, aluminum chloride, potassium chloride, sodium acetate, and petroleum ether purchased from Sigma Aldrich (MilliporeSigma, Steinheim, Germany) were used for the analysis.

### 2.2. Plant Material 

The pumpkin (*Cucurbita maxima* L.) and quince (*Cydonia oblonga* Mill.) were acquired from a supermarket in Galati, Romania in September 2021.

In total, 5 kg of each type of fruit (pumpkin and quince) was purchased from the same batch for the experiment. The selected fruits were at full maturity.

The samples were washed in order to remove the impurities and then peeled, the seeds and the stub were removed, and the flesh was cut into cubes. The samples were frozen at −15 °C for 24 h in a household freezer.

### 2.3. Puree Making

The frozen flesh cubes were mixed with 30% volume of water for 10 min at 1000 rpm with a Philips HR2100/40 blender, EC (European Community). Then, the mixtures were conventionally cooked in a Multicooker (Philips HD3037/70, 980 W, 5 L, Eindhoven, the Netherlands) using a special program for puree manufacturing (95 °C for 20 min). The final purees were packed into glass jars and stored at room temperature (21 ± 2 °C). Four samples of puree were obtained in different variants: pumpkin puree, quince puree, pumpkin and quince in a ratio of 1:1, and pumpkin and quince in a ratio of 3:1.

The purees were encoded as it follows: P—pumpkin puree, Q—quince puree, PQ 1:1—pumpkin quince puree in a ratio of 1:1, and PQ 3:1—pumpkin and quince puree in a ratio of 3:1.

### 2.4. Phytochemical Profile 

The phytochemical profile was determined by analyzing the main compounds with bioactive potential present in all the samples.

#### 2.4.1. Extraction of Bioactive Compounds 

The extraction of bioactive compounds was developed using the method of Nistor et al. [12]. All the assays were developed in triplicate.

All the puree samples were in wet form, but the reporting method used the dry weight in the calculation formula, determined by drying in an IR balance.

#### 2.4.2. Determination of Total Phenolic Content

First, 500 μL of the extract from each sample was introduced into test tubes and mixed with 2.5 mL of tenfold diluted Folin–Ciocâlteu reagent and 2 mL of 7.5% sodium carbonate. The tubes were covered with aluminum foil and allowed to stand for 30 min at room temperature before the absorbance was read at 765 nm using a UV–Vis spectrophotometer (Biochrom Libra S22 UV/Vis, Cambridge, UK) [13]. Each assay was performed in triplicate. The results were expressed as milligrams of gallic acid equivalent per gram dry weight (mg GAE/g DW) 

#### 2.4.3. Total Flavonoid Content

The aluminum chloride colorimetric method described by Marinova et al. [14] was used to determine total flavonoid content. Quercetin was used to make the standard calibration curve, and it was diluted in methanol in the range of 5–200 μg/mL. A volume of 0.6 mL of diluted standard quercetin solutions or extracts was separately mixed with 0.6 mL of 2% aluminum chloride. After mixing, the solution was incubated for 60 min at room temperature. The absorbance of the reaction mixtures was measured against blank at 420 nm wavelength with a UV–Vis spectrophotometer (Biochrom Libra S22 UV/Vis, Cambridge, UK). The concentration of total flavonoid content in the test samples was calculated from the calibration plot and expressed as mg of quercetin equivalent (QE)/g DW.

#### 2.4.4. ABTS Radical-Scavenging Assay 

The method described by Xu et al. [15] was used to determine the inhibition of ABTS^+^ radicals. The ABTS radical cation (ABTS·^+^) was produced by reacting equal volumes of 7 mM ABTS stock solution with 2.45 mM K_2_S_2_O_8_; the mixture was kept in the dark for 16 h before use. Aliquots of 1 mL of ABTS·^+^ solution were diluted with 35 mL of methanol to get an absorbance of 1.12 ± 0.02 at 734 nm. A volume of 2.85 mL of the ABTS·^+^ solution and 0.15 mL of the sample reacted for 2 h in a dark room before measuring the absorbance at 734 nm. The ABTS·+ antioxidant activity of the samples was expressed as mM Trolox equivalent/g DW according to the calibration curve. The inhibition was calculated as the ABTS radical-scavenging activity.
(1)$$(\%)=\frac{Abs_{control-Abs_{sample}}}{Abs_{control}},$$
where *Abs_control_* is the absorbance of ABTS radicals in methanol, and *Abs_sample_* is the absorbance of the ABTS radical solution mixed with sample extract/standard. All determinations were performed in triplicate.

#### 2.4.5. Extraction of Carotenoids from Pumpkin and Quince Puree 

First, 2 g of puree was homogenized with 10 mL of petroleum ether. Then, the extraction was further performed by ultrasonication for 30 min (MRC Scientific Instruments, Essex, United Kingdom). The ultrasonic bath was equipped with a digital control system of sonication time, temperature, and frequency. The extraction was performed at a constant frequency of 40 kHz and power of 100 W. The resulting supernatant was collected and centrifuged at 9000× *g*, at 10 °C for 10 min. 

#### 2.4.6. Carotenoid Content 

Carotenoid content, in terms of total carotenoids, β-carotene, and lycopene, was determined using a spectrophotometric method. The absorbance was measured at 470 nm, 450 nm, and 503 nm. The amount of carotenoids was calculated according to Souza et al. [16].
Carotenoids (mg/g) = A·M_w_·D_f_/(M_a_·L),(2)
where A is the absorbance of the petroleum ether phase at the corresponding wavelength, M_w_ is the molecular weight (536.9), Df is the sample dilution rate, M_a_ is the molar absorptivity (2500 L·mol^−1^·cm^−1^, 2590 L·mol^−1^·cm^−1^, and 3450 L·mol^−1^·cm^−1^, respectively), and L is the cell diameter of the spectrophotometer (1 cm).

#### 2.4.7. Anthocyanin Content

To determine the anthocyanin content, a volume of 3 mL of extract was diluted in 5 mL of two different buffers: 0.025 M potassium chloride, pH = 1.0, and 0.4 M sodium acetate, pH = 4.5. After 30 min of incubation at room temperature, absorption (A) was measured at λ = 510 and 700 nm. All extracts were analyzed in triplicate. 

For the calculation of total anthocyanins as cyanidin glucoside equivalent (CGE), the molar absorptivity coefficient (ε) values 26,900 M^−1^·cm^−1^ and the molecular weight of 449 Da were used. The results were calculated similarly to Giusti and Wrolstad [17] as follows: Asp = (A510 − A700) pH1.0 − (A510 − A700) pH 4.5.

The content of total anthocyanin (TA) content was calculated as follows: TA = (Asp × M × Df × 1000)/(ε × λ × m),(3)
where Df is the dilution factor, λ is the cuvette optical path length (1 cm), m is the weight of the sample in grams, and Asp is the aspirin control group.

The total anthocyanin content was expressed as mg CGE/g DW.

### 2.5. Water Activity

A LabSwift water activity instrument (Novasina, Lachen, Switzerland) was used to measure the water activity (a_w_) of the purees.

### 2.6. Texture Analysis 

Texture is important for fruit purees because it induces the specific sensorial attributes appreciated by consumers. The texture profile analysis (TPA) method was performed with a Brookfield CT3 Texture Analyzer. The method consisted of a double penetration of a 25.4 mm diameter acrylic cylinder, to a depth of 10 mm, into the samples packed into cylindric plastic containers (30 mm diameter and 43 mm height). The penetration speed was 1 mm/s, and the trigger load was 0.067 N. TexturePro CT V1.5 software was used to collect the parameter values. Firmness, adhesiveness, cohesiveness, and springiness were determined. The presented results are the means of the five values.

### 2.7. Rheological Analysis

The rheology of puree samples was determined using the equipment and method described by Nistor et al. [18].

### 2.8. Color Measurements 

Surface color of the fresh and processed puree samples was recorded using a MINOLTA Chroma Meter CR-410 (Konica Minolta, Osaka, Japan). Color properties were evaluated using the CIE (*L**, *a**, *b**) system. The total color difference (Δ*E*), color intensity (chroma, *C**), visual color appearance (hue angle, *h**), browning index (*BI*), and yellowness index (*YI*) were calculated according to Equations (4)–(8), respectively [19].
(4)$$∆E=\sqrt{{\left(L_{0}^{*}-L^{*}\right)}^{2}+{\left(a_{0}^{*}-a^{*}\right)}^{2}+{\left(b_{0}^{*}-b^{*}\right)}^{2}},$$
where subscript 0 refers to the color of the fresh sample.
(5)$$C^{*}=\sqrt{a^{*2}+b^{*2}}.$$
(6)$$h^{*}=tan^{-1}\left(\frac{b^{*}}{a^{*}}\right).$$
(7)$$BI=100\times \left(\frac{X-0.31}{0.17}\right),$$
where X=(a*+1.75·L*)(5.645·L*+a*−3.012·b^).
(8)$$\;YI=\frac{142.86\cdot b^{*}}{L^{*}}.$$

The color parameters are dimensionless. The color analyses were performed in triplicate.

### 2.9. Confocal Laser Scanning Microscopy (CLSM)

To highlight the main structural changes of the puree samples, a CLSM was applied. For this study, we used the confocal laser system and the same method described by Nistor et al. [18].

### 2.10. Sensorial Analysis

The sensorial properties of the puree samples were analyzed by 20 panelists from Food Science and Engineering Faculty staff. General aspect, smooth aspect, specific taste, sweetness intensity, specific color, flavor, consistency, adhesiveness, spreadability, and acceptability were evaluated on a five-point scale (1—extremely dislike, 5—extremely like) [20]. Samples were presented to be evaluated in transparent, colorless plastic containers, each of them provided with a three-digit code. Bread and drinking water were offered to the panelists to clear their palates between samples tasting. Analysis of variance (ANOVA) was used to determine differences between puree samples at a significance level of 5%. For consistency and adhesiveness, the linear regression for instrumental and sensorial tests was applied, in order to determine the correlation between data.

### 2.11. Data Analysis

For each puree sample, the results are reported as the mean ± standard deviation (SD). The software Minitab 19.0 (free trial) was used to evaluate the statistical differences between the puree samples. In this regard, one-way analysis of variance (ANOVA) and Tukey’s test with a 95% confidence interval was applied when significant differences were observed. Principal component analysis (PCA) was used to estimate the relationships between phytochemical and color parameters of different puree samples. The standardized data were presented in a matrix of four lines (samples) and 15 columns (phytochemical and color parameters), and then the PCA method was used.

## 3. Results and Discussions 

### 3.1. Phytochemical Profile of the Purees

The phytochemical profile in terms of β-carotene, lycopene, and TPC was determined for the four samples, and the water activity was also measured.

Table 1 presents the results for the phytochemical profile of the puree samples.

Regarding the control samples treated only by conventional cooking, it can be seen that the highest value of the total carotenoids could be attributed to P_0_, followed by PQ 3:1, as expected due to the high content of pumpkin.

Compared to the control samples (P_0_ and Q_0_), freezing had a positive impact on the puree samples; an increase of 5–14% in TPC was obtained for the puree samples, while TFC registered an increase of almost 10% for both pumpkin and quince purees.

Similar findings were reported by [20] for several methods of drying including freeze-drying.

It can be seen that the combination of freezing and conventional cooking for a short time could increase carotenoid content, as a consequence of better extraction due to the exposure of the cellular content.

As expected, the highest values of the β-carotene content were obtained by the samples containing mainly pumpkin, such as P (5.34 ± 0.05 mg/g DW) and PQ 3:1 (3.78 ± 0.014 mg/g DW). The values presented in Table 1 for β-carotene content are 10 times higher than those obtained by Becker et al. [21] for sterilized pumpkin puree. In the case of quince puree, a higher value of 0.66 ± 0.001 mg/g DW was determined compared to that reported by Legua et al. [22] for nine varieties of quince (from 0.04 to 0.42 mg/100 g). The same tendency was revealed for the lycopene content. Thus, the highest value of 0.84 ± 0.01 mg/g DW for the lycopene content was attributed to the pumpkin puree. Even in its raw form, quince has no red compounds; after processing, it turns into a reddish color, explained by the release of anthocyanin from the cells. These are sustained by the anthocyanin content, which registered the highest value for quince puree (0.61 ± 0.02 mg C3G/g DW).

Pumpkin puree is one of the richest products in total phenolic content, and it seems that the processing did not affect these valuable compounds, since the registered value was 47.54 ± 0.1 mg GAE/g, which is 40% higher than the values for pumpkin jam presented by Zdunić et al. [23]

The total phenolic content registered the highest value for quince puree (68 ± 0.2 mg GAE/g DW), in accordance with Ponder and Hallmann [24], who reported the results for *Chaenomeles* and quince. 

The water activity (a_w_) is a very important parameter involved in food preservation, which indicates the main chemical interactions of the hydrophilic compounds. Table 1 presents the a_w_ in pumpkin and quince puree samples processed by freezing and cooking at 95 °C for 20 min. No significant differences (*p* < 0.05) were determined for the four purees samples. The water activity has an important role in solid food compounds; thus, in canned or similar products, it must be above 0.85 (FDA, 2014) [25]. The quince puree registered the highest value for water activity (0.874 ± 0.02) due to its compact structure. For the other types of purees, the water activity ranged between 0.865 and 0.872, indicating the destruction of the vegetative cells of microorganisms and the spoilage microbiota. With respect to this affirmation, the samples had the proper characteristics of preserved foods.

### 3.2. The Results of ABTS Radical-Scavenging Activity

The results of the ABTS radical-scavenging activity are presented in Figure 1.

Many studies have revealed the importance of processing some parts of the plants such as aerial parts, roots, stems, flowers, and fruits to increase the bioavailability of the most important bioactive compounds. The pumpkin and quince antioxidant activity are recognized for their beneficial effects on cardiovascular diseases and hypertension [24,26]. Figure 1 shows the antioxidant properties of pumpkin and quince fruit puree samples. All samples were found in a good position above 71% quenching of the ABTS cation radicals to stop the oxidation process by accepting an electron. The highest value (76.25% ± 0.26%) of ABTS inhibition was registered by PQ 3:1, while the lowest value was attributed to the P sample (71.32% ± 0.31%). It can be said that the addition of quince puree increased the antioxidant activity of the combined puree.

### 3.3. Texture Profile Analysis Results

The results of the texture determinations are presented in Table 2.

Firmness was determined as the maximum penetration force registered for the first cycle. A firmness between 1.27 ± 0.11 and 2.33 ± 0.19 N was determined by pumpkin puree, while the combined puree variants registered intermediate values. These results are due to the content of pectin in quince, which creates firm gels [27]. Since pumpkin contains lower levels of pectin comparing to quince, it induces a weaker texture of the puree samples. When the pumpkin puree was predominant in sample (PQ 3:1), the values of the firmness were close to the values of P sample. These values are in accordance with those reported by Garrido et al. [28] for apple jelly. At the same time, the firmness values are slightly smaller compared to those reported by Curi et al. [27] for quince marmalades, as a function of the product formulation (the cited authors used quince juice, which has a lower content of dry matter, compared to pulp).

Adhesiveness represents the work required to pull the probe away from the sample [27]. As in the case of firmness, the adhesiveness of the combined samples registered values (8.34 ± 0.18 mJ for PQ 1:1 sample and 6.17 ± 0.23 N for PQ 3:1 sample) between the individual sample values. This behavior could be associated with the factors influencing adhesiveness, such as the humidity of the samples [27]. 

Cohesiveness represents the strength of the internal bonds making up the body of the product [26]. It registered values between 0.37 ± 0.02 and 0.63 ± 0.03. These values depend on the puree microstructure and could be compared with those reported by Curi et al. [27] for quince marmalades.

Springiness is a measure of the puree elasticity. It is expressed as the capacity of the samples to recover the deformation. From Table 2, it can be noted that quince samples showed the lowest springiness value (5.32 ± 0.18 mm), while pumpkin samples showed the highest value (8.42 ± 0.11 mm). The other samples registered values between them.

### 3.4. Rheological Measurements Results

The results of rheological measurements are expressed in Figure 2.

The dynamic rheological properties of pumpkin and quince purees, as well as their combinations, are shown in Figure 2. 

All the puree samples showed a decrease in G′ and G″ with increasing frequency, which indicates that the elasticity and viscosity correspond to the change in apparent viscosity. 

This kind of behavior could be attributed to the formation of intermolecular aggregates facilitated by an increase in the polysaccharide or biopolymer concentration. 

For the passion fruit pulp with the addition of some hydrocolloids, Moraes et al. [29] obtained comparable results with the present study.

The values of both moduli were slightly dependent on the frequency and strain; the values of G′ exceeded those of G″ in the frequency range of 1–100 Hz, whereby G′ increased from 3173 to 8749 Pa, while G″ increased from 539.5 to 2336 Pa for pumpkin puree. These data show that the purees behaved like a weak gel [30] as observed for tomato products and Indian coffee pulp [31]. The presence of native pectin could influence the rheology of the samples in accordance with the results reported by Martínez et al. [32] who used butternut squash peel as a source of pectin in the obtention of papaya jam.

### 3.5. Color Measurements Results

The results for the color measurements are presented in Table 3.

Color represents an important parameter considered fundamental by the consumers for the visual appreciation of purees. The color stability of the samples can be affected by various factors such as oxygen, pH, heat temperature, light and sugar content, the presence of ascorbic acid or some metal ions, and the amount of pigment present in fresh product and thermal processed purees [33,34]. Through the puree manufacturing process, color modification cannot be avoided due to the formation of browning and yellowing compounds, Maillard reactions, and polymerization reactions of anthocyanins with some phenolic compounds [35].

The values of *L**, which show the whiteness of the product, fluctuated between 67.28 ± 0.27 and 76.54 ± 0.16 for fresh samples and between 44.72 ± 0.16 and 54.86 ± 0.25 for processed samples. The thermal treatment caused a decrease in the *L** value between 18.46% and 37.82% for the processed samples when compared to the control samples. The purees were characterized by a darkened color determined by various factors such as processing temperature and time, type of the cultivars, vitamin C content, and some pigments found in the raw material. The *L** values established for pumpkin purees, quince purees, and mixed fruit (pumpkin with quince) purees were similar to those presented by Wojdylo et al. [36] for strawberry jam, Rababah et al. [37] for cherry jam, and Banaś et al. [38] for gooseberry jam.

In the fresh samples, analyzed immediately after making, the value of the *a** parameter ranged between 1.88 ± 0.06 and 18.56 ± 0.35. Table 3 shows an increase in the *a** values after heat treatment in puree samples compared with the control samples. Similar findings were reported by Cascales and García [39] for commercial jam obtained from Spanish quince.

The *b** values of puree samples decreased by 10.84% and 43.31% when compared with the values of the fresh samples. The lowest value was found in mixed fruit puree (pumpkin with quince in proportion 1:1) (26.62 ± 0.41); on the other hand, the highest value of this color parameter was recorded for the puree obtained only from pumpkin (43.59 ± 0.05), which is a natural result of the color of this raw material. Chutintrasri and Noomhorm [40], who analyzed pineapple puree, also found a decrease in the *b** parameter during thermal treatment. ΔE of the puree samples varied from 24.23 ± 0.19 to 34.89 ± 0.15 units, influenced by the type of raw material, the applied combination of freezing and thermal treatment, or the processing time.

From Table 3, it can be observed that the values of *C** (color saturation) and *h** (hue angle) for the puree samples decreased by 2.37–34.51% and 20.12–26.89%, respectively. The values of these color parameters also depend on processing variables (temperature and time) and possibly the type of cultivar.

The *L**, *a**, and *b** values were used to calculate the yellowness index (*YI*) and browning index (*BI*), indicating the purity of the brown color. *YI* is related to product deterioration by light exposure and processing. The increased *YI* and *BI* values for puree obtained only from quince and mixed fruit samples (pumpkin with quince in proportion 3:1) compared to the fresh samples could be assigned to Maillard reactions or pigment release.

### 3.6. Confocal Laser Scanning Microscopy (CLSM) Images

The results of confocal laser scanning microscopy (CLSM) are presented in Figure 3.

After processing, the pumpkin and quince purees preserved almost 80% of the original tissue structures. Thus, in Figure 3a,b, parenchymal cells (from pumpkin) can be observed specific to the mesocarp, while prosenchymatic cells (from quince) can also be seen, filled with the specific bio compounds such as carotenoids (C), which emitted green fluorescence in the 500–600 nm range. The pumpkin cells were characterized by almost the same dimensions evidenced in Figure 3a (l = 124.44 μm/L = 164 μm) and (l = 102.25 μm/L = 133.24 μm), while quince (b) presented some elongated cells with l = 196.14 μm/L = 59.08 μm and l = 138.12 μm/L = 73.23 μm. These findings are similar to De Carvalho et al. [41] for *Cucurbita moschata*.

In the purees of pumpkin with quince (1:1 Figure 3c and 1:3 Figure 3d), there were both parenchymal (l = 212.92 μm/L = 105.74 μm) and prosenchymatic cells (l = 256.58 μm/49.43 μm) belonging to each type of fruit. The maintenance of cell integrity was due to the combination of treatments freezing and soft processing, which preserved the content of the cells represented by dissolved sugars, acids, polyphenols, water-soluble color substances, and inorganic compounds [41].

### 3.7. Sensorial Analysis

The average scores given by the panelists were used to construct the radar diagram for sensorial analysis (Figure 4).

For the general aspect, the most appreciated sample was that containing only quince. The smooth aspect gained the best score for pumpkin puree, while the lowest score was obtained for quince puree. The mixed samples received intermediate values. These results could be explained by the microstructure of the fruits; the parenchymal cells in the pumpkin induced a smooth aspect, while the prosenchymatic cells in quince induced a rough aspect. When combining the purees, the smooth aspect of the quince puree was enhanced. For the sweetness intensity, the lowest scores were obtained by the quince puree, due to the organic acids contained in the fruits. PQ 1:1 was perceived as the sweetest sample, probably because of the mutual potentiation of organic acids from quince and sugars from pumpkin. For the specific color, the panelists considered that P and PQ 3:1 were the most attractive samples. The consistency of the samples was related to their cohesiveness. When comparing the results of instrumental and sensorial texture analysis, the linear correlation coefficient values were 0.9682 for consistency and 0.9733 for adhesiveness. For the overall acceptability, appropriate values of the scores were obtained, showing that the combination of quince and pumpkin could be a suitable solution for obtaining valuable fruit purees. ANOVA analysis showed no significant differences (*p* < 0.05) between scores given by panelists. 

### 3.8. Principal Component Analysis 

The PCA plots (Figure 5A,B) explained the relationship between puree samples and the phytochemical and color parameters assessed in this study.

The location of puree formulations is presented in Figure 5A, and the distribution of phytochemical and color parameters in space as defined by the first and second PCA dimensions is shown in Figure 5B. The sum of principal components PC1 and PC2 explained 95% of the variation among pumpkin, quince, and mixed fruit (pumpkin and quince in different proportions) puree samples. PCA reduced the number of variables into two principal components, with components 1 and 2 explaining 70.8% and 24.2% variability, respectively. From the results presented in Figure 5A, it can be seen that the samples encoded P and PQ 3:1 were located in the same quadrant; hence, it can be concluded that there were no significant differences between these samples. Since Q was in a different quadrant from the other puree samples, it exhibited a qualitative significant difference with the other samples. Similarly, the thermally processed puree sample (PQ 1:1) was the only sample in the third quadrant.

The PCA plot (Figure 5B) showed a cluster of color parameters (*b**, *C**, *a**, *YI*, and *BI*) with some bioactive compounds (*β*-carotene and lycopene). Another cluster (the second) included some antioxidant compounds (TPC, TFC, and anthocyanins) grouped together with the *h** (hue angle) and total color change (Δ*E*). On the basis of this plot, it is possible to certify that samples P and PQ 3:1 had the highest bioactive compound content. Comparable observations could be concluded in the case of Q and PQ 1:1 puree samples for TPC, TFC, and anthocyanin content. The PCA results confirmed the correlation analysis between phytochemical and color parameters in the case of fruit puree samples.

## 4. Conclusions 

This study provided important information on the characteristics of pumpkin and quince puree, which could be applied in the food industry. 

It can be concluded that pumpkin and quince processing did not diminish the antioxidant quality of these valuable fruits. When combining pumpkin with quince puree, the complex matrix evidenced the benefits of both fruits. The color, textural, rheological, structural, and sensorial characteristics suggest the suitability of these purees to be consumed as functional foods.

Through processing of pumpkin and quince, the shelf life may be increased, and post-harvest losses can be minimized.

## Figures and Tables

**Figure 1 foods-11-02038-f001:**
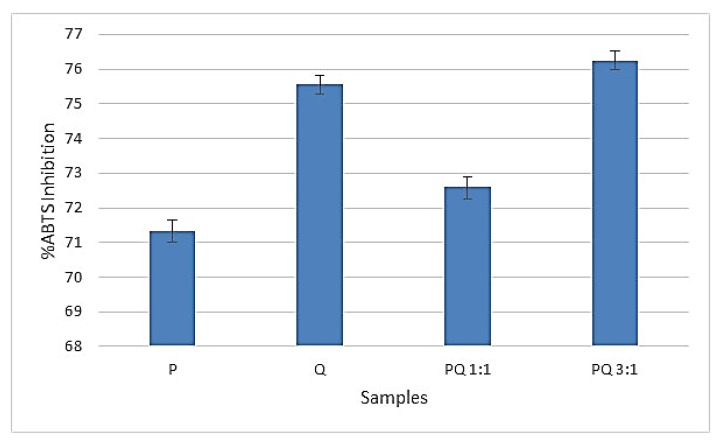
The inhibition of ABTS by the antioxidant activity of pumpkin and quince puree.

**Figure 2 foods-11-02038-f002:**
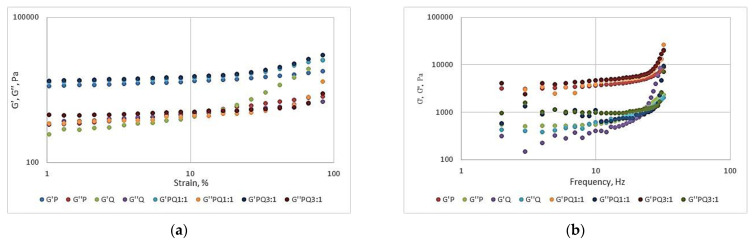
Elastic modulus (G′) and viscous modulus (G″) variation with strain and frequency: (**a**) the elastic modulus (G′) and viscous modulus (G″) versus strain; (**b**)the elastic modulus (G′) and viscous modulus (G″) versus frequency. P—pumpkin puree, Q—quince puree, PQ 1:1—pumpkin and quince puree in a ratio of 1:1, PQ 3:1—pumpkin and quince puree in a ratio of 3:1.

**Figure 3 foods-11-02038-f003:**
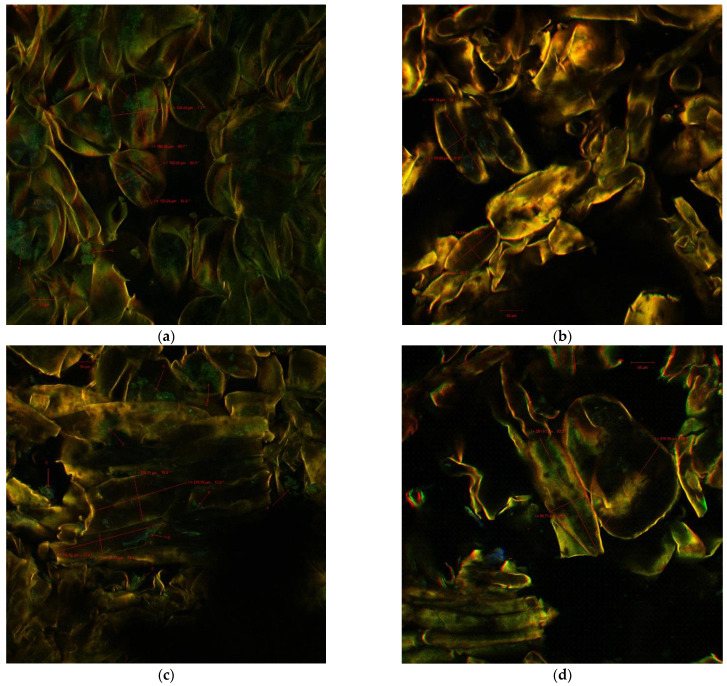
The CLSM images for the puree samples: (**a**) pumpkin puree; (**b**) quince puree; (**c**) pumpkin and quince puree 1:1; (**d**) pumpkin and quince puree 3:1.

**Figure 4 foods-11-02038-f004:**
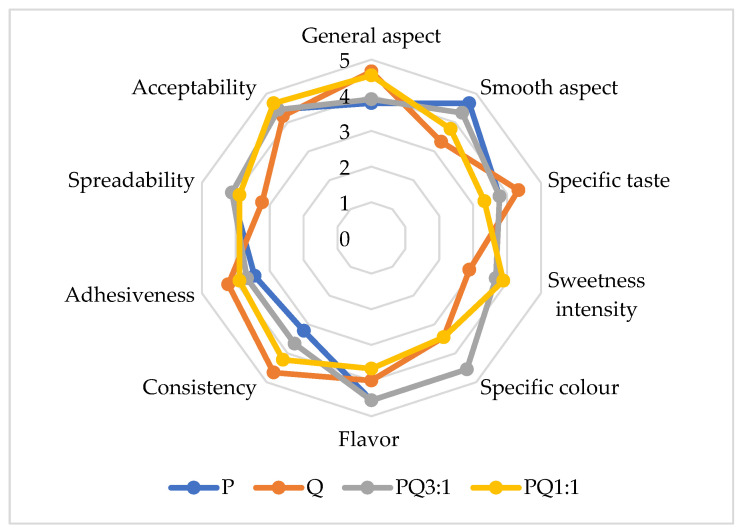
Radar diagram for sensorial analysis.

**Figure 5 foods-11-02038-f005:**
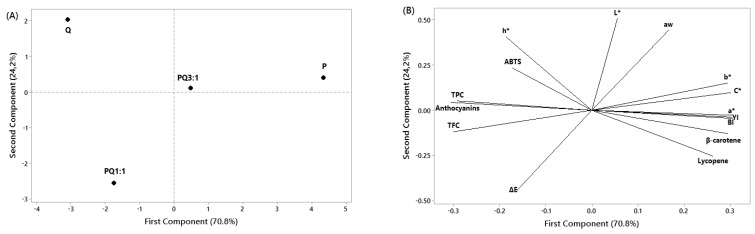
Principle component analysis (PCA) of fruit puree: (**A**) the location of different samples; (**B**) the location of phytochemical and color parameters.

**Table 1 foods-11-02038-t001:** Phytochemical profile of pumpkin and quince puree samples.

Samples	P_0_	P	Q_0_	Q	PQ 1:1	PQ 3:1
Total carotenoids, mg/g DW	5.4 ± 0.1 ^A^	6.14 ± 0.1 ^A^	0.65 ± 0.02 ^A^	0.7 ± 0.01 ^B^	3 ± 0.03 ^A^	4.54 ± 0.1 ^A^
β-carotene, mg/g DW	4.44 ± 0.02 ^C^	5.34 ± 0.05 ^C^	0.60 ± 0.02 ^C^	0.66 ± 0.001 ^C^	2.62 ± 0.02 ^C^	3.78 ± 0.014 ^C^
Lycopene, mg/g DW	0.74 ± 0.03 ^C^	0.84 ± 0.01 ^C^	-	-	0.55 ± 0.03 ^C^	0.65 ± 0.01 ^C^
Anthocyanins, mg CGE/g DW	-	-	0.58 ± 0.01 ^B^	0.61 ± 0.02 ^C^	0.45 ± 0.02 ^C^	0.27 ± 0.02 ^C^
TPC, mg GAE/g DW	45.25 ± 0.2 ^A^	47.54 ± 0.1 ^A^	60 ± 0.2 ^A^	68 ± 0.2 ^A^	62 ± 0.3 ^A^	52 ± 0.2 ^A^
TFC, mg/100 g DW	23 ± 0.15 ^B^	25.26 ± 0.05 ^B^	44 ± 0.1 ^B^	48.40 ± 0.01 ^B^	50.05 ± 0.05 ^B^	38 ± 0.1 ^B^
a_w_, %	0.8 ± 0.07 ^C^	0.872 ± 0.01 ^C^	0.8 ± 0.02 ^C^	0.874 ± 0.02 ^C^	0.865 ± 0.03 ^C^	0.871 ± 0.01 ^C^

TPC, total phenolic content; TFC, total flavonoid content; TAC, total anthocyanin content; a_w_, water activity; P_0_—conventionally cooked pumpkin puree, Q_0_—conventionally cooked quince puree, P—pumpkin puree, Q—quince puree, PQ 1:1—pumpkin and quince puree in 1:1 ratio, PQ 3:1—pumpkin and quince puree in 3:1 ratio. Values are represented as the mean ± standard errors. Significant differences between the samples are evidenced by superscript letters (A, B, C) with *p* > 0.05.

**Table 2 foods-11-02038-t002:** Textural parameters of pumpkin and quince purees.

Sample	P	PQ	PQ 1:1	PQ 3:1
Firmness, N	1.27 ± 0.11 ^B^	2.33 ± 0.19 ^B^	1.98 ± 0.17 ^B^	1.51 ± 0.09 ^B^
Adhesiveness, mJ	5.49 ± 0.28 ^A^	10.21 ± 0.46 ^A^	8.34 ± 0.18 ^A^	6.17 ± 0.23 ^A^
Cohesiveness, -	0.37 ± 0.02 ^B^	0.63 ± 0.03 ^B^	0.52 ± 0.02 ^B^	0.47 ± 0.03 ^B^
Springiness, mm	8.42 ± 0.11 ^A^	5.32 ± 0.18 ^A^	6.08 ± 0.09 ^A^	7.57 ± 0.14 ^A^

Values are represented as the mean ± standard errors. Different superscript letters (A, B) indicate a significant difference at (*p* > 0.05) among different samples.

**Table 3 foods-11-02038-t003:** Color attributes of fresh and thermal processed purees.

Parameters Sample Code	*L**	*A**	*B**	Δ*E*	*C**	*H**	*BI*	*YI*
*Fresh* (control) *samples*								
P_0_	67.28 ± 0.27 ^A^	18.56 ± 0.35 ^A^	61.52 ± 0.39 ^A^	–	64.26 ± 0.23 ^A^	73.22 ± 0.39 ^A^	158.53 ± 0.41 ^A^	130.62 ± 0.26 ^A^
Q_0_	76.54 ± 0.16 ^B^	−1.88 ± 0.06 ^B^	32.39 ± 0.11 ^B^	–	32.45 ± 0.12 ^B^	– 86.69 ± 0.11 ^B^	15.88 ± 0.00 ^B^	60.46 ± 0.33 ^B^
P_0_ Q_0_ 1:1	71.92 ± 0.06 ^A,B^	8.34 ± 0.14 ^A,B^	46.96 ± 0.11 ^A,B^	–	47.70 ± 0.09 ^A,B^	79.93 ± 0.19 ^A,B^	71.47 ± 0.41 ^A,B^	93.29 ± 0.30 ^A,B^
P_0_ Q_0_ 3:1	73.42 ± 0.07 ^A,B^	9.84 ± 0.15 ^A,B^	48.46 ± 0.12 ^A,B^	–	49.45 ± 0.08 ^A,B^	78.52 ± 0.18 ^A,B^	74.42 ± 0.42 ^A,B^	94.31 ± 0.29 ^A,B^
*Thermal processed purees*								
P	53.18 ± 0.03 ^B^	26.72 ± 0.07 ^C^	43.59 ± 0.05 ^B,C^	24.23 ± 0.19 ^B,C^	51.13 ± 0.02 ^B,C^	58.49 ± 0.10 ^A,B^	142.55 ± 0.34 ^A^	117.10 ± 0.14 ^A^
Q	54.86 ± 0.25 ^B^	13.01 ± 0.06 ^C^	28.88 ± 0.28 ^B,C^	26.54 ± 0.26 ^B,C^	31.68 ± 0.28 ^B,C^	65.75 ± 0.11 ^A,B^	54.51 ± 0.34 ^A^	75.22 ± 0.39 ^A^
PQ 1:1	44.72 ± 0.16 ^B^	16.35 ± 0.28 ^C^	26.62 ± 0.41 ^B,C^	34.89 ± 0.15 ^B,C^	31.24 ± 0.44 ^B,C^	58.44 ± 0.43 ^A,B^	77.65 ± 0.68 ^A^	85.06 ± 0.61 ^A^
PQ 3:1	49.66 ± 0.32 ^B^	18.34 ± 0.25 ^C^	34.53 ± 0,14 ^B,C^	28.86 ± 0.25 ^B,C^	39.10 ± 0.08 ^B,C^	62.03 ± 0.22 ^A,B^	100.19 ± 0.22 ^A^	99.36 ± 0.29 ^A^

Values are represented as the mean ± standard errors. Different superscript letters (A, B, C.) mindicateean a significant difference between the samples. P_0_—fresh pumpkin puree, Q_0_—fresh quince puree, P_0_Q_0_ 1:1—fresh pumpkin and quince puree in a ratio of 1:1, P_0_Q_0_ 3:1—fresh pumpkin and quince puree in a ratio of 3:1, P—thermal processed pumpkin puree, Q—thermal processed quince puree, PQ 1:1—thermal processed pumpkin and quince puree in a ratio of 1:1, PQ 3:1—thermal processed pumpkin and quince puree in a ratio of 3:1.

## Data Availability

The data presented in this study are available on request from the corresponding author. The data are not publicly available due to privacy.

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
