# Peer review of "A Complex Characterization of Pumpkin and Quince Purees Obtained by a Combination of Freezing and Conventional Cooking"

_foods, 2022, doi:10.3390/foods11142038_

Round 1
Reviewer 1 Report
I have reviewed the manuscript entitled: A complex characterization of pumpkin and quince pu-2 rees obtained by a combination of freezing and conventional cooking.
The biggest reservations is that the Authors presented the novelty of the study as the ‘combination of technologies based on freezing and thermal processing’ and concluded that these ‘combination of preserving treatment has a positive impact on the nutritional value and the content of bioactive compounds’, but there was the lack of any control sample for example treated with the one of these preserving treatment (e.g. only thermal treatment without freezing). The is no basis for such conclusion in the experiment conducted and planned in such way.
Moreover, the Authors concluded that ‘thermal at 95 ÌŠC for 20 min was an optimal combination, although they did not test any others.
Moreover, the is the lack of the description of the results in the Results section.
Such descriptions were given in the section of Discussion. I suggest to move these descriptions to Result section or combine both section together in Results and Discussion.
Specific comments:
Abstract, line 20 – rather ‘pumpkin puree’ instead of ‘apple puree’
Line 134 – Was the samples used in wet or dried form for the extraction? The results e.g. of total phenolic or antioxidant activity were expressed per g dw (dry weight). Was the dry matter determined? If yes, it should be mentioned in the methods.
Line 195 – Please, explain the abbreviation ‘Asp’
Line 201 - the abbreviation aw should be provided in this section. It was used later in the text in the section of Discussions.
Lines 270 -274 – This phrase should be move to section of Data analysis.
Results – Please complete the results descriptions or combine with the Discussions.
Line 437 and 453 – Please, do not start a sentence with an abbreviation. It will be better write ‘The L*’ and ‘The b*’ or ‘The value of L*’
Lines 484 – 487 – Please, fill in any Reference
Author Response
Title: A complex characterization of pumpkin and quince purees obtained by a combination of freezing and conventional cooking
Manuscript Number: foods-1790878
The authors would like to thank for the suggestions and the spent time to review the manuscript. We are hopping that the answers are suitable to change your opinion and to clarify some scientifical aspects.
Reviewer 1#
Comment:
The biggest reservations is that the Authors presented the novelty of the study as the ‘combination of technologies based on freezing and thermal processing’ and concluded that these ‘combination of preserving treatment has a positive impact on the nutritional value and the content of bioactive compounds’, but there was the lack of any control sample for example treated with the one of these preserving treatment (e.g. only thermal treatment without freezing). The is no basis for such conclusion in the experiment conducted and planned in such way.
Answer:
We understand your vision, but the authors have considered that the simple samples (P and Q) could constitute the control samples, while for the other ones could be determined the possible interactions. Moreover, the results reported to other findings have registered similar or higher results, so we have considered that the combination of freezing and thermal processing is beneficial for our products.
Comment:
Moreover, the Authors concluded that ‘thermal at 95 ÌŠC for 20 min was an optimal combination, although they did not test any others.
Answer:
Other tests for determining the processing time were developed. The authors concluded this in terms of the results related to the obtained samples, as a conclusion for the potential of these treatments to be used for other vegetal materials with similar good results.
Comment:
Moreover, the is the lack of the description of the results in the Results section.
Such descriptions were given in the section of Discussion. I suggest to move these descriptions to Result section or combine both sections together in Results and Discussion.
Answer:
As you have suggested to give more scientifical meaning to the results we have merged the results and discussion in one part
Specific comments
Comment:
Abstract, line 20 – rather ‘pumpkin puree’ instead of ‘apple puree’
Answer:
We have operated the replacing.
Comment:
Line 134 – Was the samples used in wet or dried form for the extraction? The results e.g. of total phenolic or antioxidant activity were expressed per g dw (dry weight). Was the dry matter determined? If yes, it should be mentioned in the methods.
Answer:
The samples were in wet form, but the reporting method suppose the using of the dry weight in the calculation formula, determined by drying in an IR balance.
Comment:
Line 195 – Please, explain the abbreviation ‘Asp’
Answer:
Asp-Aspirin control group
Comment:
Line 201 - the abbreviation aw should be provided in this section. It was used later in the text in the section of Discussions.
Answer:
We have added the abbreviation.
Comment:
Lines 270 -274 – This phrase should be move to section of Data analysis.
Answer:
We have operated the modification of these sections.
Comment:
Results – Please complete the results descriptions or combine with the Discussions.
Answer:
We have completed the Results chapter with discussions.
Comment:
Line 437 and 453 – Please, do not start a sentence with an abbreviation. It will be better write ‘The L*’ and ‘The b*’ or ‘The value of L*
Answer:
We have operated the suggestions.
Comment:
Lines 484 – 487 – Please, fill in any Reference
Answer:
We have operated the suggestions.
Reviewer 2 Report
Abstract and introduction are well written, except the novelty refer to cited reference WHO, 2015!
Material and method part
Extraction of Bioactive Compounds! it is better to use sample extraction
and need to recheck your subjects.
the results are very poor written. All data are presented in tables and figures as follows:
In Table 1 are presented the results for the phytochemical profile of the puree samples!
The results of the ABTS radical scavening activity are presented in figure 1!
All data in the results are presented in the same way. Which is not scientifically acceptable.
If the author prefers to merge results and discussion in one part will be more logic
Author Response
Title: A complex characterization of pumpkin and quince purees obtained by a combination of freezing and conventional cooking
Manuscript Number: foods-1790878
The authors would like to thank for the suggestions and the spent time to review the manuscript. We are hopping that the answers are suitable to change your opinion and to clarify some scientifical aspects.
Reviewer 2#
Comment:
Abstract and introduction are well written, except the novelty refer to cited reference WHO, 2015!
Answer:
We have substituted the reference: The importance and novelty of the study is sustained by the accordance with the international regulations regarding the reducing of the sugar intake among adults and children [9]
9. Centers for Diseases Control and Prevention (CDC) https://www.cdc.gov/nutrition/data-statistics/added-sugars.html/24.06.2022/06.29pm
Comment:
Material and method part
Extraction of Bioactive Compounds! it is better to use sample extraction and need to recheck your subjects.
Answer:
For all the determinations of the bioactive compounds were used alcoholic extracts as it was described in the article at the subchapter 2.4.1.
Comment:
the results are very poor written. All data are presented in tables and figures as follows:
In Table 1 are presented the results for the phytochemical profile of the puree samples!
The results of the ABTS radical scavening activity are presented in figure 1!
All data in the results are presented in the same way. Which is not scientifically acceptable.
Answer:
I understand your suggestion, but indifferent what results we have to reconsider the presentation will be the same, table or columns/bars. Because we have a comparation between samples, not a variety depending on time, which could be more feasible for a linear graph. Moreover, in the article are also images from CLSM, a radar graph for sensorial analysis and linear graphs for statistical interpretation.
Comment:
If the author prefers to merge results and discussion in one part will be more logic
Answer:
To give more scientifical meaning to the results we have merged the results and discussion in one part as you have suggested.
Reviewer 3 Report
The manuscript is interesting but it should be careful reviewed by the authors.
Line 14: correct Pumpkin (Cucurbita maxima) in Pumpkin (Cucurbita maxima L.) and Quince (Cydonia oblonga) in Quince (Cydonia oblonga Mill.)
Line 20: apple puree?
Line 60: refer to literature data on lycopene content in pumpkin. In general, as regard the carotenoid content pumpkin, the following paper should be considered: Durante M. et al. 2014 Supercritical Carbon Dioxide Extraction of Carotenoids from Pumpkin (Cucurbita spp.): A Review. International Journal of Molecular Sciences, 15, 6725-6740.
Line 89: add a reference
Line 150: report the details of the total flavonoid content evaluation
Line 189: report the molecular weight considered to evaluate the total carotenoid content
Line 289: Why did the authors evaluate the content of lycopene and not of lutein or xanthophylls? Considering that after alpha and beta carotene, lutein is the most carotenoid present in high content while lycopene is almost absent or in any case present in low quantitiy compared to the other carotenoids.
Furthermore, I would suggest to consider not only beta carotene but also alpha carotene and then to change beta carotene in the sum alpha plus beta carotene.
Line 349: reference number 17 is referred to cherry fruits. Check!
Line 380: check the information report in reference number 22
Author Response
Title: A complex characterization of pumpkin and quince purees obtained by a combination of freezing and conventional cooking
Manuscript Number: foods-1790878
The authors would like to thank for the suggestions and the spent time to review the manuscript. We are hopping that the answers are suitable to change your opinion and to clarify some scientifical aspects.
Reviewer 3#
Comment:
Line 14: correct Pumpkin (Cucurbita maxima) in Pumpkin (Cucurbita maxima L.) and Quince (Cydonia oblonga) in Quince (Cydonia oblonga Mill.)
Answer:
Thank you! We have operated the request.
Comment:
Line 20: apple puree?
Answer:
Thank you! The error was corrected with pumpkin puree.
Comment:
Line 60: refer to literature data on lycopene content in pumpkin. In general, as regard the carotenoid content pumpkin, the following paper should be considered: Durante M. et al. 2014 Supercritical Carbon Dioxide Extraction of Carotenoids from Pumpkin (Cucurbita spp.): A Review. International Journal of Molecular Sciences, 15, 6725-6740.
Answer:
We have operated your suggestion.
Comment:
Line 89: add a reference
Answer:
We have operated the suggestion.
Comment:
Line 150: report the details of the total flavonoid content evaluation
Answer:
Quercetin was used to make the standard calibration curve and it was diluted in methanol in the range of 5–200 μg/mL. A volume of 0.6 mL of diluted standard quercetin solutions or extracts was separately mixed with 0.6 mL of 2% aluminum chloride. After mixing, the solution was incubated for 60 min at room temperature. The absorbance of the reaction mixtures was measured against blank at 420nm wavelength with a UV–Vis spectrophotometer (Biochrom Libra S22 UV/Vis, Cambridge, United Kingdom). The concentration of total flavonoid content in the test samples was calculated from the calibration plot and expressed as mg quercetin equivalent (QE)/g dw.
Comment:
Line 189: report the molecular weight considered to evaluate the total carotenoid content
Answer:
The molecular weight of the carotenoids is 536.9.
Comment:
Line 289: Why did the authors evaluate the content of lycopene and not of lutein or xanthophylls? Considering that after alpha and beta carotene, lutein is the most carotenoid present in high content while lycopene is almost absent or in any case present in low quantitiy compared to the other carotenoids.
Furthermore, I would suggest to consider not only beta carotene but also alpha carotene and then to change beta carotene in the sum alpha plus beta carotene.
Answer:
Thank you for your suggestion, for sure we will take it into consideration your recommendation for further studies, but for the present one it will be hard to do the determination of lutein or alpha carotene because we don’t have anymore the fruits or the purees from the same batch, so we risk to report different values. Even the season does not afford the opportunity to obtain these fruits, because right now are just in the growing stage.
Comment:
Line 349: reference number 17 is referred to cherry fruits. Check!
Answer:
We have changed the reference with Nistor, O.V., Bolea, C.A., Andronoiu, D.G., Cotârlet, M., Stănciuc, N. Attempts for Developing Novel Sugar-Based and Sugar-Free Sea Buckthorn Marmalades. Molecules, 2021. 26, 1-10, 3073. https://doi.org/10.3390/molecules26113073.
Comment:
Line 380: check the information report in reference number 22
Answer:
Thank you! We have operated the request.
Reviewer 4 Report
the manuscript describes the effects of cooking and freezing of pumpkin and quince. The article is discreet even if at times a little confused in the organization of tables and figures.
Materials and methods need to be improved: line 112: the choice of samples and sampling needs to be detailed. How many samples were purchased? Has a pool of products been made from the various stores?
Enter details of the cooking carried out, and if it can be defined as conventional as it is done in the title. How were cooking times and temperatures established? are there any references about it?
Author Response
Title: A complex characterization of pumpkin and quince purees obtained by a combination of freezing and conventional cooking
Manuscript Number: foods-1790878
Reviewer 4#
the manuscript describes the effects of cooking and freezing of pumpkin and quince. The article is discreet even if at times a little confused in the organization of tables and figures.
Comment:
Materials and methods need to be improved: line 112: the choice of samples and sampling needs to be detailed. How many samples were purchased?
Answer:
5 kg of each type of fruit, apple and respectively quince from the same batch were purchased for the experiment. The selected fruits were at full maturity.
Comment:
Has a pool of products been made from the various stores?
Answer:
The fruits were purchased from only one market and from one single batch. The study does not follow a comparation between the varieties of fruits sold in different stores.
Comment:
Enter details of the cooking carried out, and if it can be defined as conventional as it is done in the title.
Answer:
Then, the mixtures were conventional cooked in a Multicooker (Philips HD3037/70, 980 W, 5 L, Eindhoven, the Netherlands) at a special program for puree manufacturing (95°C for 20 min).
For the manufacturing of purees, it was used an electronic pot a Multicooker (Philips HD3037/70, 980 W, 5 L, which does not need any additional procedures to cook by itself the purees, but the program setting.
Comment:
How were cooking times and temperatures established? are there any references about it?
Answer:
The cooking temperature is a default value of the PUREE PROGRAM of the Multicooker.
Round 2
Reviewer 1 Report
Title: A complex characterization of pumpkin and quince purees obtained by a combination of freezing and conventional cooking
Manuscript Number: foods-1790878
The authors would like to thank for the suggestions and the spent time to review the manuscript. We are hopping that the answers are suitable to change your opinion and to clarify some scientifical aspects.
General comment:
In my opinion, the manuscript could be accepted for publication only when the main aim of the study and conclusions will be change, so that they match the results. I think that the results present the study on the impact of mixing two ingredients P and Q on the tested parameters. The work does not present the results proving the impact of combining treatments (freezing and cooking) on the puree quality as the Authors maintain in the description of novelty in the Introduction and in the Conclusions, because there is the lack of the control. It causes that the results of the study are incompatible with the aim and the Conclusion.The Authors should clearly define the aim, which will be accordance with obtained results. In my opinion, it should be impact of the mixtures Q and P on quality of obtained purees. The conclusions should be revised.
Reviewer 1#
Comment:
The biggest reservations is that the Authors presented the novelty of the study as the ‘combination of technologies based on freezing and thermal processing’ and concluded that these ‘combination of preserving treatment has a positive impact on the nutritional value and the content of bioactive compounds’, but there was the lack of any control sample for example treated with the one of these preserving treatment (e.g. only thermal treatment without freezing). The is no basis for such conclusion in the experiment conducted and planned in such way.
Answer:
We understand your vision, but the authors have considered that the simple samples (P and Q) could constitute the control samples, while for the other ones could be determined the possible interactions. Moreover, the results reported to other findings have registered similar or higher results, so we have considered that the combination of freezing and thermal processing is beneficial for our products.
Comment to answer:
It seems to me that the Authors did not fully understand my objection.
‘Simple samples (P and Q)’ could be a control for tested mixtures, but still they all were treated in the same way (freezing and cooking), so there is no basic for conclusion that combined treatment by cooking and freezing is beneficial more than for example only cooking treatment.
I still think that there should be ‘no freeze control’ as this was presented as the main novelty at work. The authors want to demonstrate the benefit of combining two treatment methods (cooking and freezing). How can this be achieved without examining the traditional approach (only cooking treatment)? How do we know if cooking alone would not be a better approach after all?
I consider two conclusions of the work to be unjustified:
“The results showed that the combination of preserving treatments has a positive impact on the nutritional value and the content of bioactive compounds in the pumpkin and quince puree samples.”
and
“Processing in general qualitatively and quantitatively could alter some important biological compounds, but it this case the coupling of freezing and thermal treatment at 95°C for 20 min was an optimum combination.”
In fact, the authors studied only the influence of mixing pumpkin and quince. All results present the comparison of single ingredients (P and Q) and they mixtures. So, I can agree with the last conclusion, which actually concerns the results of this work:
“When combining pumpkin with quince puree the complex matrix has evidenced the benefits of both fruits. The colour, textural, rheological, structural and sensorial characteristics recommend these purees to be consumed as functional foods.”
Comment:
Moreover, the Authors concluded that ‘thermal at 95 ÌŠC for 20 min was an optimal combination, although they did not test any others.
Answer:
Other tests for determining the processing time were developed. The authors concluded this in terms of the results related to the obtained samples, as a conclusion for the potential of these treatments to be used for other vegetal materials with similar good results.
Comment to answer:
The Authors did not optimize the temperature and time. The conclusion is unfounded.
Comment:
Moreover, there is the lack of the description of the results in the Results section.
Such descriptions were given in the section of Discussion. I suggest to move these descriptions to Result section or combine both sections together in Results and Discussion.
Answer:
As you have suggested to give more scientifical meaning to the results we have merged the results and discussion in one part
Comment to answer:
OK
Specific comments
Comment:
Abstract, line 20 – rather ‘pumpkin puree’ instead of ‘apple puree’
Answer:
We have operated the replacing.
Comment to answer:
Ok
Comment:
Line 134 – Was the samples used in wet or dried form for the extraction? The results e.g. of total phenolic or antioxidant activity were expressed per g dw (dry weight). Was the dry matter determined? If yes, it should be mentioned in the methods.
Answer:
The samples were in wet form, but the reporting method suppose the using of the dry weight in the calculation formula, determined by drying in an IR balance.
Comment to answer:
Please, write this information in the Methods.
Comment:
Line 195 – Please, explain the abbreviation ‘Asp’
Answer:
Asp-Aspirin control group
Comment to answer:
OK
Comment:
Line 201 - the abbreviation aw should be provided in this section. It was used later in the text in the section of Discussions.
Answer:
We have added the abbreviation.
Comment to answer:
OK
Comment:
Lines 270 -274 – This phrase should be move to section of Data analysis.
Answer:
We have operated the modification of these sections.
Comment to answer:
OK
Comment:
Results – Please complete the results descriptions or combine with the Discussions.
Answer:
We have completed the Results chapter with discussions.
Comment to answer:
OK
Comment:
Line 437 and 453 – Please, do not start a sentence with an abbreviation. It will be better write ‘The L*’ and ‘The b*’ or ‘The value of L*
Answer:
We have operated the suggestions.
Comment to answer:
OK
Comment:
Lines 484 – 487 – Please, fill in any Reference
Answer:
We have operated the suggestions.
Comment to answer:
OK
Author Response
Title: A complex characterization of pumpkin and quince purees obtained by a combination of freezing and conventional cooking
Manuscript Number: foods-1790878
The authors would like to thank the reviewers for the close reading and for the proper suggestions and comments purposed to improve the scientific quality of the paper. The present version of the manuscript has been revised and completed with the specific points which have been addressed by the referents.
Reviewer 1#
Comment:
The biggest reservations is that the Authors presented the novelty of the study as the ‘combination of technologies based on freezing and thermal processing’ and concluded that these ‘combination of preserving treatment has a positive impact on the nutritional value and the content of bioactive compounds’, but there was the lack of any control sample for example treated with the one of these preserving treatment (e.g. only thermal treatment without freezing). The is no basis for such conclusion in the experiment conducted and planned in such way.
Answer:
We understand your vision, but the authors have considered that the simple samples (P and Q) could constitute the control samples, while for the other ones could be determined the possible interactions. Moreover, the results reported to other findings have registered similar or higher results, so we have considered that the combination of freezing and thermal processing is beneficial for our products.
Comment to answer:
It seems to me that the Authors did not fully understand my objection.
‘Simple samples (P and Q)’ could be a control for tested mixtures, but still they all were treated in the same way (freezing and cooking), so there is no basic for conclusion that combined treatment by cooking and freezing is beneficial more than for example only cooking treatment.
I still think that there should be ‘no freeze control’ as this was presented as the main novelty at work. The authors want to demonstrate the benefit of combining two treatment methods (cooking and freezing). How can this be achieved without examining the traditional approach (only cooking treatment)? How do we know if cooking alone would not be a better approach after all?
Answer:
We have completed with your suggestion.
Table 1. Phytochemical profile of pumpkin and quince puree samples
Samples |
P0 |
P |
Q0 |
Q |
PQ1:1 |
PQ3:1 |
Total carotenoids, mg/g DW |
5.4±0.1A |
6.14±0.1A |
0.65±0.02A |
0.7±0.01B |
3±0.03A |
4.54±0.1A |
β-carotene, mg/g DW |
4.44±0.02C |
5.34±0.05C |
0.60±0.02C |
0.66±0.001C |
2.62±0.02 C |
3.78±0.014 C |
Lycopene, mg/g DW |
0.74±0.03C |
0.84±0.01C |
- |
- |
0.55±0.03 C |
0.65±0.01 C |
Anthocyanins, mg CGE/g dw |
- |
- |
0.58±0.01B |
0.61±0.02 C |
0.45±0.02 C |
0.27±0.02 C |
TPC, mg GAE/g DW |
45.25±0.2 A |
47.54±0.1 A |
60±0.2A |
68±0.2 A |
62±0.3 A |
52±0.2 A |
TFC, mg/ 100 g DW |
23±0.15 B |
25.26±0.05 B |
44±0.1B |
48.40±0.01B |
50.05±0.05 B |
38±0.1 B |
aw, % |
0.8±0.07 C |
0.872±0.01 C |
0.8±0.02 C |
0.874±0.02 C |
0.865±0.03 C |
0.871±0.01 C |
TPC, total phenolic content; TFC, total flavonoid content; TAC, total anthocyanin content; aw, water activity; P0- pumpkin puree and Q0 - conventional cooked quince puree, P-pumpkin puree, Q, quince puree, PQ1:1, pumpkin and quince puree in 1:1 ratio, PQ3:1, pumpkin and quince puree in 3:1 ratio. Values are represented as mean ± standard errors. The significant differences between the samples are evidenced by superscript letters (a, b, c) with p >0.05.
Regarding the control samples treated only by conventional cooking, it could be seen that the highest value of the total carotenoids is attributed to P0, followed by PQ3:1, which is expected due to the high content of pumpkin.
Compared to the control samples (P0 and Q0), freeze has a positive impact on the puree samples, an increase of 5-14% in TPC was obtained for the puree samples while TFC has registered an increase of almost 10% both for pumpkin and quince purees.
Similar findings were reported by [20] for several methods of drying including freeze-drying.
It could be seen that the combination between freezing and conventional cooking for short time, can increase carotenoid content, being a consequence of better extraction due to the exposure of the cellular content.
Comment:
I consider two conclusions of the work to be unjustified:
“The results showed that the combination of preserving treatments has a positive impact on the nutritional value and the content of bioactive compounds in the pumpkin and quince puree samples.”
and
“Processing in general qualitatively and quantitatively could alter some important biological compounds, but it this case the coupling of freezing and thermal treatment at 95°C for 20 min was an optimum combination.”
In fact, the authors studied only the influence of mixing pumpkin and quince. All results present the comparison of single ingredients (P and Q) and they mixtures. So, I can agree with the last conclusion, which actually concerns the results of this work:
“When combining pumpkin with quince puree the complex matrix has evidenced the benefits of both fruits. The colour, textural, rheological, structural and sensorial characteristics recommend these purees to be consumed as functional foods.”
Answer:
We have deleted the conclusions that you do not consider proper.
Comment:
Moreover, the Authors concluded that ‘thermal at 95 ÌŠC for 20 min was an optimal combination, although they did not test any others.
Answer:
Other tests for determining the processing time were developed. The authors concluded this in terms of the results related to the obtained samples, as a conclusion for the potential of these treatments to be used for other vegetal materials with similar good results.
Comment to answer:
The Authors did not optimize the temperature and time. The conclusion is unfounded.
Answer:
We have deleted the conclusion regarding the temperature and time.
Reviewer 2 Report
Thank you for your efforts to revise the manuscript.
part 3 is still results ! and as you mentioned you have merged results and discussion
the results still need to rewrite to be more understanding for the readers
Author Response
Title: A complex characterization of pumpkin and quince purees obtained by a combination of freezing and conventional cooking
Manuscript Number: foods-1790878
The authors would like to thank the reviewers for the close reading and for the proper suggestions and comments purposed to improve the scientific quality of the paper. The present version of the manuscript has been revised and completed with the specific points which have been addressed by the referents.
Reviewer 2#
Comment:
Part 3 is still results ! and as you mentioned you have merged results and discussion
The results still need to rewrite to be more understanding for the readers
Answer:
Thank you! We have operated the suggestion.
Reviewer 3 Report
Since the analysis of total carotenoids has been reported in the materials and methods section (line 180). The autors should be report these values in the table to establish the contribution of β carotene and lycopene content on the total carotenoids content.
In this regard I again suggest to consider the following paper: Durante M. et al. 2014 Supercritical Carbon Dioxide Extraction of Carotenoids from Pumpkin (Cucurbita spp.): A Review. International Journal of Molecular Sciences, 15, 6725-6740.
Furthermore, the reference number 25 is the same of the reference number 4.
Author Response
Title: A complex characterization of pumpkin and quince purees obtained by a combination of freezing and conventional cooking
Manuscript Number: foods-1790878
The authors would like to thank the reviewers for the close reading and for the proper suggestions and comments purposed to improve the scientific quality of the paper. The present version of the manuscript has been revised and completed with the specific points which have been addressed by the referents.
Reviewer 3#
Comment:
Since the analysis of total carotenoids has been reported in the materials and methods section (line 180). The autors should be report these values in the table to establish the contribution of β-carotene and lycopene content on the total carotenoids content.
In this regard I again suggest to consider the following paper: Durante M. et al. 2014 Supercritical Carbon Dioxide Extraction of Carotenoids from Pumpkin (Cucurbita spp.): A Review. International Journal of Molecular Sciences, 15, 6725-6740.
Answer:
We have presented and commented the values for total carotenoids, as you have suggested. We hope, this time is ok.
Comment:
Furthermore, the reference number 25 is the same of the reference number 4.
Answer:
Thank you! We have changed the reference with
- Montesano, D., Rocchetti, G., Putnik, P., Lucini, L., Bioactive profile of pumpkin: an overview on terpenoids and their health-promoting properties. Current Opinion in Food Science 2018, 22, 81–87, https://doi.org/10.1016/j.cofs.2018.02.003